# ‘It’s Easily the Lowest I’ve Ever, Ever Got to’: A Qualitative Study of Young Adults’ Social Isolation during the COVID-19 Lockdowns in the UK

**DOI:** 10.3390/ijerph182211777

**Published:** 2021-11-10

**Authors:** Chloe C. Dedryver, Cécile Knai

**Affiliations:** Faculty of Public Health and Policy, London School of Hygiene and Tropical Medicine, London WC1H 9SH, UK; cecile.knai@lshtm.ac.uk

**Keywords:** mental health, COVID-19, social isolation, young people, anxiety, depression, pandemic, psychological impact

## Abstract

(1) Background: Social connectivity is key to young people’s mental health. Local assets facilitate social connection, but were largely inaccessible during the pandemic. This study consequently investigates the social isolation of young adults and their use of local assets during the COVID-19 lockdowns in the UK. (2) Methods: Fifteen semi-structured Zoom interviews were undertaken with adults aged 18–24 in the UK. Recruitment took place remotely, and transcripts were coded and analysed thematically. (3) Results: Digital assets were key to young people’s social connectivity, but their use was associated with stress, increased screen time and negative mental health outcomes. The lockdowns impacted social capital, with young people’s key peripheral networks being lost, yet close friendships being strengthened. Finally, young people’s mental health was greatly affected by the isolation, but few sought help, mostly out of a desire to not overburden the NHS. (4) Conclusions: This study highlights the extent of the impact of the pandemic isolation on young people’s social capital and mental health. Post-pandemic strategies targeting mental health system strengthening, social isolation and help-seeking behaviours are recommended.

## 1. Introduction

Social connectivity is key to young people’s health [1]. For the purpose of this study, social connectivity refers to perceived connections with others, from family, to friends, to work colleagues, which are valued, supportive, and mutual [2]. Social connectivity enables social capital, which is the sum of resources (physical or virtual) accrued through reciprocal relationships [3]. Social capital has a protective effect for young people’s wellbeing [4], as do strong friendships [5]. Individual social capital can promote prosocial behaviours and reduce difficult behaviours in youth [6]. Moreover, friendships help develop social competencies, and provide important support to young people during challenging life experiences [7]. Strong social networks are also correlated with positive development, and improved self-esteem, happiness, and school adjustment [7].

Conversely, social isolation is an important risk factor for poor mental health [5]. The World Health Organisation (WHO) defines mental health as a state of well-being where an individual can cope with stress, work productively and realise their abilities [8]. Mental health conditions usually emerge during youth (12–24), with half of all disorders beginning by age 14 [9,10]. Poor mental health is associated with broader health concerns for young people, such as substance abuse, lower educational attainment, poor reproductive and sexual health, and violence [10]. Moreover, there are clear links between poor mental health and unhealthy diet, especially for young adults [1,11].

The COVID-19 lockdowns had important implications for mental health. Sudden and significant changes in routine have led to higher anxiety levels, irrational fears, and panic attacks [12]. The ongoing uncertainty surrounding the pandemic, coupled with political ambiguity, have also contributed to heightened anxiety, notably due to the possible mortality risk [13]. Indeed, fears around infecting the self or others are expected to lead to an increase in post-traumatic stress disorder (PTSD) and depression [12]. Moreover, the disproportionate allocation of media time and occasionally sensationalist covering of the pandemic in the news has been underlined as a risk factor for anxiety [14].

Indeed, in 2020, nearly a fifth (19%) of all adults in the United Kingdom (UK) suffered from depression, representing more than 8 million people [15]. This is nearly twice the pre-pandemic depression levels (10%) [15]. Anxiety and depression prevalence among young (18–24) UK adults during the lockdowns have been abnormally high; in 2021, young adults (16–29) were nearly five times more likely to suffer from depression compared to other age groups [15,16]. Locating the study in the UK is relevant beyond the scope of the lockdowns, moreover, as anxiety and depression levels have been on the rise for young people (16–24) in the UK since 2017, especially for young women [17]. Poor mental health was already the primary cause of disability-adjusted life-years for people aged 15–49 in the UK before the pandemic [18].

Local assets are key to enabling the development of social ties. Here, these refer to resources, physical or not, which contribute to life in the community and facilitate connectedness. Existing research underlines how physical spaces contribute to social connectivity by functioning as meeting places for young people. Examples in the UK include parks, pubs, restaurants and gyms. For instance, parks promote social cohesion by enabling the mingling of different ethnic groups [19]. These assets can also be intangible. A valued community service, for example, such as a cultural organisation, can contribute to social cohesion, as can key community residents, by facilitating local events.

However, local assets can have a range of public health impacts which sometimes compete. For example, in the UK, chicken shops are important community spaces, and serve as a meeting place for adolescents [20,21]. These are ubiquitous fast-food outlets selling deep-fried chicken pieces and chips (fries) which are rooted in urban youth culture [21]. While these places contribute to social connectivity, through the food consumed there, they also contribute to obesity [22]. Yet, during the lockdowns, many local assets promoting social capital amongst young people shut. As a result, young adults resorted to new ways of developing social connections. However, little is known about what these assets are, and what the impact of their use is for public health.

Existing research examines the impact of the pandemic on young people’s mental health, but the studies are largely quantitative [23,24,25]. Literature highlights the role of social isolation, and lack of social support, as key risk factors for mental disorders during the lockdowns [26,27]. Moreover, the role of local assets in facilitating social capital amongst young people is underlined [28,29]. However, no studies to our knowledge bridge the gap between the role of local assets in young people’s social isolation during the COVID-19 pandemic, and assess the subsequent impacts on mental health.

This study consequently aims to investigate the social isolation of young people in England and their use of local assets during the COVID-19 lockdowns. The qualitative nature of the study enabled an in-depth exploration of young UK adults’ lived experiences of the lockdowns, generating insights into their attitudes, values and perceptions of the effects of the pandemic. The objectives are: (1) investigate the changing use and role of local assets in social connectivity for young people in the UK during the lockdowns; and (2) generate insights into the impact of the lockdowns on the lived experiences of young people in the UK, primarily in terms of social isolation and mental health.

## 2. Materials and Methods

The study background, rationale and topic guide (Appendix A) were informed by structured literature searches of Medline and PsycInfo, and by snowballing the references of relevant articles. Keyword and subject heading searches were conducted for variations on the terms “asset*” OR “friend*” OR “young people” OR “COVID*” OR “uk”.

The study included youth as defined by the WHO: individuals aged 15–24, but excluded those under the age of 18 for ethical considerations [30]. Inclusion criteria were to have lived in the UK during lockdowns, have access to a computer for remote interviews, and be aged 18–24. The focus on this age group is particularly important as they were one of the age ranges most impacted by the lockdowns.

Recruitment took place remotely due to the pandemic. A range of strategies were employed, including: cold-emailing and calling youth hubs and organisations; approaching mental health charities, such as Mind; reaching out to online support groups for mental health on Facebook; posting on social media platforms, including Twitter and Instagram; asking personal contacts to circulate the call for participants.

Following written consent, semi-structured Zoom interviews were conducted and recorded, following the themes in the topic guide. This was informed by an initial literature search of Medline and PsychInfo on spaces employed by young people during the pandemic for social connectivity in the UK. The topic guide was piloted with a personal contact prior to data collection, ensuring the questions were comprehensive, fluid and unambiguous. The semi-structured nature of the interviews also enabled participants to draw attention to new topics, and provide a personal account of their experiences [31]. The topic guide was refined on an ongoing basis, including additional probes according to the data generated [31]. Participants were sent a help sheet following the interviews, with further resources for mental health support.

Data analysis started at the outset of the project, as interviews were transcribed and coded while the data collection was ongoing. The data were examined thematically, according to Braun and Clarke’s (2006) sixfold approach [32]. First, the interviews were transcribed, with attention paid to non-verbal cues. Next, the data were coded manually, generating inductive initial codes in a systematic line-by-line fashion. The codes were then reviewed, and collated into potential themes. This involved both a top-down approach, where the themes were informed by the literature searches and objectives, as well as a bottom-up approach, which drew out the themes inductively from the data. Next, the themes were refined, reviewed, and named, ensuring that they accurately reflected the dataset. Finally, data extracts were selected which appropriately portrayed the sub-themes, labelled by participant number (e.g., P01).

## 3. Results

### 3.1. Respondent Characteristics

A total of fifteen participants were recruited (Table 1). Interviews lasted between 26 and 57 min. Eleven participants were female, and four were male, with ages ranging from 18 to 24 (mean = 22). Respondents included seven university and A-level students, three part-time employees, two private sector workers, a graduate trainee, a mental health professional and an unemployed participant, spanning from middle to high socioeconomic status (SES). Of the fifteen respondents, six had pre-existing mental health conditions, and two (P09, P12) indicated suffering from symptoms of mental health conditions despite not having been formally diagnosed. Participants were based in England and Wales, primarily in North and North West England, and London.

Respondents’ narratives drew attention to the role of digital assets in maintaining social connectivity during the UK lockdowns, but their use was simultaneously associated with negative health outcomes (Table 2). Online spaces, such as Zoom, were key in recreating normalcy and connecting young people, but were also reported as isolating, leading to reply anxiety, stress and increased screen time. Despite engaging in digital interactions, participants reported a loss of social capital and peripheral friendships, counterbalanced by the strengthening of inner circle relationships. Finally, respondents underlined the impact of the enforced social isolation on their mental health. All reported that their mental wellbeing had deteriorated over the last year to varying degrees, regardless of diagnosis status. However, few sought help, primarily out of a desire to not overburden the NHS.

### 3.2. Local Assets for Connectivity

To counter the social isolation brought about by the UK lockdowns, participants reported resorting to online platforms to maintain their social connections (e.g., Zoom, Houseparty, FaceTime). These were cited as enabling a recreation of normalcy, which was key in keeping up respondents’ support networks.


*‘I mean, I can’t imagine going through something like this, like the pandemic, and everything being closed and not being able to see anyone, without kind of the technology that we have available...because even though it’s not the same and it’s not, it’s not quite as good, it’s close enough. And you can still feel like you can you can connect with people.’—P10*


However, these spaces were paradoxically described as isolating, draining, and an unsatisfactory substitute to real-life interactions.


*‘I wish I could see them and I’d much prefer to do something else. So in that sense it was a bit sad and slightly isolating, I guess, because it’s all behind a screen.’—P15*



*‘some people just don’t like video calls. And I totally get that because it can be quite exhausting. I think it is draining.’—P03*


Moreover, respondents underlined disliking, and feeling guilty about, the screen time associated with online platform use. Maintaining friendships online also triggered a stress response and reply anxiety amongst participants.


*‘And then then I got guilt because, like, I went on my screen time and my screen time was like nine hours a day. And I’m like, oh, my days like, I can’t be doing this. That’s like, my brain’s gonna die—and it’s not, but you know. Like then I felt guilty about spending so much time on my phone.’—P08*



*‘I’m really shit at texting and social media. It kind of gives me anxiety. I got really bad like reply anxiety, where if someone messages me, feel like I need to reply instantly and then I end up resenting them because they keep messaging me.’—P13*


### 3.3. Loss of Social Capital

Both the enforced isolation and the stress accompanying digital interactions entailed a loss of social capital amongst respondents during the lockdowns. Participants highlighted the strain of maintaining of online friendships. Respondents also reported feeling uncomfortable, observed, and on show during their digital interactions.


*‘it’s just exhausting trying to keep up friendships like online sometimes.’—P04*



*‘at some points it was kind of like a long distance relationship.’—P15*



*‘I’d say that was probably the only unhelpful aspect, is just seeing myself on camera 24/7 and kind of like analysing flaws, et cetera.’—P14*


Participants’ broader friendship networks were greatly affected. Respondents underlined finding it increasingly difficult to maintain group friendships and dynamics. Furthermore, interviewees reported growing apart from friends on the peripheries of their social circle, but who were nonetheless important in their daily social lives.


*‘I think it definitely made us as a friendship group drift more from each other, which was difficult.’—P01*



*‘it’s hard to like gather everybody at the same time… we’re a little bit more like disjointed now, so like the like group, like it’s hard to sort of find where it begins and ends now.’—P04*



*‘it just cut me off from quite a lot of people who... So it wasn’t like really good friends who I had, but who I was just seeing and just seeing at college every day’—P02*


However, despite the largely detrimental impact of the lockdowns on social connectivity, the pandemic also pushed young people to get closer with their inner social circle. Participants reported making more effort to stay in touch with their close friends, and learning key organisational and communication skills to this effect. Interviewees also highlighted having more honest and sensitive conversations with friends, underlined as an important support mechanism for respondents’ wellbeing.


*‘I think some of them have got like a lot closer, actually, because you had to be so intentional about, like maintaining friendships in lockdown.’—P08*



*‘I’d probably say that we’ve checked in more in terms of mental health, so I found myself asking friends, how are you doing, as in, really how are you doing? And I think that’s sort of been reciprocated as well, which has been really good.’—P03*


### 3.4. Worsening Mental Health

Nonetheless, the impact of the lockdowns on young people’s social connectivity and daily lives significantly affected mental health. All respondents reported that their mental wellbeing had deteriorated since March 2020 to varying degrees. Participants without a clinical diagnosis of a mental health condition underlined feeling overwhelmed and crying more than usual.


*‘I’ve had days where I’ve kind of opened my laptop and just—sounds really silly—but just cried because I’ve just been like I literally just cannot cope’—P03*


On the other hand, participants with a formal diagnosis indicated that their mental health had been considerably aggravated. Two interviewees (P01, P13) disclosed suicidal ideation.


*‘The first lockdown was absolutely appalling. It was awful. That was the worst I’ve been in years. I relapsed really, really badly with my eating disorder’—P14*



*‘there was a point where I thought I was... I was suicidal and it’s easily the lowest I’ve ever, ever got to, the lowest point.’—P13*


Worryingly, few participants reported seeking help. Those with formal diagnoses often stated intentionally not getting support, out of a desire to not burden friends, family, or the National Health Service.


*Everybody was spiralling. And I felt really bad for taking up people’s time and space when they were struggling too. I did not want to feel like a burden.’—P14*


Respondents also frequently underlined finding the ongoing sense of uncertainty during the pandemic challenging. This was reported as contributing to poor mental health, and triggering anxiety spirals, leaving participants feeling *‘quite panicky’* (P04). Moreover, respondents indicated fearing coronavirus, notably during the first lockdown, leading to compulsive safety behaviours, such as washing hands *‘so, so, so much’* (P12). Finally, interviewees highlighted the challenge of managing an unstable timetable for relaxing lockdown restrictions. This was a source of concern which participants reported as affecting their mindset and outlook.


*‘not knowing how long it would be before I could, like, see my friends or, like, hug my grandparents, like was obviously really difficult when you already feel, when when I already felt like life was a bit pointless and I struggled a lot with what I was doing with my life anyway, that obviously like intensified it and made it a lot harder to think positively about the future.’—P10*


## 4. Discussion

This study sought to investigate the social isolation of young UK adults and their use of local assets during the COVID-19 lockdowns. Assets most employed to maintain social connectivity amongst this sample were digital platforms. Online spaces such as social media have been reported as playing a crucial role in young people’s social connectedness and experiences of the lockdowns beyond the scope of the UK [33,34]. These have been found to contribute to young people’s wellbeing by providing a necessary space for social interaction and everyday communication [33,34]. However, although online platforms promise connection, they can paradoxically leave users feeling drained, isolated and negatively impact mental health. Digital spaces therefore functioned not only as assets for connection, but also as potential drivers of poor mental health during the pandemic. Indeed, significant usage of digital platforms was a risk factor for anxiety and depression over the lockdowns [35,36]. Social media use amongst young people during the pandemic has resulted in experiences of emotional exhaustion, isolation, social media fatigue and screen fatigue [37]. Moreover, increased exposure to disaster news through social media has been associated with depression [38]. Beyond the scope of mental health, higher levels of screen time have been associated with binge eating during the lockdowns, and are a known risk factor for excessive weight gain [39,40].

Secondly, young people’s social connectivity was impacted by the pandemic despite the use of online platforms to maintain relationships. Young adults lost important social networks, perceiving social pressure and stress in maintaining their digital relationships, and experiencing the distancing of peripheral friendships. Existing research indicates that young people felt less friend support during COVID-19 [41]. Yet, perceived peer support is a protective factor against poor mental health [5]. Changes in social relationships have therefore been correlated with increased symptoms of depression, anxiety, and loneliness amongst young people during the pandemic [41,42]. Thus, it is recommended that local governments, in conjunction with mental health charities, implement age-appropriate and contextually tailored interventions targeting social isolation in young people. For instance, befriending interventions have been reported as cost-effective and effective both for loneliness and mental health outcomes [43,44,45,46,47]. These interventions introduce the participant to ‘one or more individuals, whose aim is to provide additional social support through the development of an affirming, emotion-focused relationship’ [47]. It is notable, however, that such interventions would benefit from face-to-face delivery, and would therefore only be appropriate when COVID-19 related social distancing restrictions do not apply.

Thirdly, young people are one of the age groups whose mental health has been the most affected by the lockdowns. The Organisation for Economic Cooperation and Development (OECD) highlights that mental health issues amongst young people doubled or more in most countries between 2020–2021 [48]. Moreover, in March 2021, young people were 30% to 80% more likely to display depressive or anxious symptoms than adults [48]. Participants reported their mental wellbeing significantly deteriorating, experiencing COVID-19-related anxiety, and struggling to manage the changing and uncertain lockdown timelines. Disruptions in health services combined with a lack of help-seeking behaviours, moreover, has meant that young adults are not accessing appropriate support [49]. Participants raised concerns around burdening the NHS by seeking mental health services during the pandemic. This echoes existing literature, which underlines that individuals with mental health disorders rarely display help-seeking behaviours [50]. Moreover, both patients and primary-care practitioners largely experienced the NHS as over-stretched during the COVID-19 lockdowns, a perception compounded by media portrayals [51]. This has ramifications for the burden of mental health disorders in the UK post-pandemic, as well as for the management of mental health services nationally. It is suggested that future qualitative studies undertake methodologically robust research into the barriers to help-seeking amongst young people, notably for those hard-to-reach [52].

The findings of this study also highlight the need to consider the long-term practical implications of the pandemic for mental health. We are likely to experience a lag between the end of containment measures and a decrease in demand on mental health services [53]. Young people have been particularly prone to increased levels of mental health issues in the months following the easing of restrictions [53]. Therefore, additional and sustained investment beyond the scope of the pandemic in psychological counselling services for young people is required [53]. Moreover, the pandemic has drawn attention to the need to strengthen mental health systems. The WHO has identified ten strategies to this effect: (a) planning for long-term sustainability from the outset; (b) addressing the population’s broad mental health needs; (c) respecting the central role of government; (d) engaging national professional organizations; (e) ensuring effective coordination across agencies; (f) reviewing mental health plans and policies as part of reform; (g) strengthening the mental health system as a whole; (h) investing in health workers; (i) using demonstration projects to raise funds for wider reform; (j) investing in advocacy to maintain momentum for change [54]. Most importantly, at every stage within this process, it is crucial that the rights of persons with mental disorders be safeguarded, both in granting equal levels of care to those infected with SARS-CoV-2, and in the provision of sufficient and appropriate services during and after the pandemic [55,56].

## 5. Limitations

The sample of participants was diverse, albeit uneven, in terms of age, gender, and geographical location. Of the 15 participants, 11 were female and 4 were male. Moreover, 14 respondents were based in England, and one in Wales (P08), but none in Scotland or Northern Ireland. Additionally, although the participants ranged from middle to high SES, none were of low SES. Given that access to a computer was an inclusion criterion for remote interviews, the portion of young UK adults who do not have access to technology would have been excluded from the study. The research consequently did not examine the differences in experiences of the COVID-19 lockdowns, despite the known inequalities in impact of the pandemic by SES [57].

Challenges of remote interviewing may also have affected the data produced, such as difficulties in gauging non-verbal cues [58], often compounded by technical issues. For example, both the researcher and P03 had to turn cameras off during the interview due to connectivity problems. Rapport building is a further challenge of remote interviewing. Nonetheless, the use of conventional techniques such as attentiveness, fluency with the topic guide and active interviewing enabled elements of remote rapport-building during the data collection process [59]. A final limitation of online interviews regards the privacy of the interview setting. Two respondents (P05, P09) were not able to carry out the interviews in private spaces. This might have affected the reliability of the data produced, as the sensitive nature of the research lent itself to a private environment.

## 6. Conclusions

This research sheds light on the extent to which the COVID-19 lockdowns affected young people’s social capital and mental health. Strategies targeting mental health system strengthening, social isolation, help-seeking behaviours, and which transcend system’s shocks such as pandemics, should be prioritised.

## Figures and Tables

**Table 1 ijerph-18-11777-t001:** Participant characteristics.

Participant Number	Age (Years)	Gender	Occupation 2020–2021	Pre-Existing Mental Health Condition (Yes/No)	Location/County
P01	23	F	Graduate scheme trainee	Yes	London
P02	19	M	Part-time supermarket worker	No	Shropshire
P03	24	F	Paralegal	No	South Yorkshire
P04	22	F	Part-time worker	Yes	North Yorkshire
P05	22	M	Student	Yes	London
P06	24	F	Private sector employee	No	London
P07	23	F	PhD candidate	No	Cornwall
P08	23	F	Part-time supermarket worker	No	South Wales
P09	21	F	Student	No	Hampshire
P10	24	F	Unemployed	Yes	West Yorkshire
P11	20	M	Student	No	London
P12	22	F	Student & frontline worker	No	London
P13	24	M	Mental health professional	Yes	Staffordshire
P14	18	F	A-level student	Yes	Worcestershire
P15	21	F	Student	No	Greater Manchester

Participant characteristics by participant number, age in years, gender, occupation during the lockdowns, and location in the UK.

**Table 2 ijerph-18-11777-t002:** Main themes.

Theme	Summary
Local assets for connectivity	Online platforms were key to young people’s social connectedness but were paradoxically described as draining and isolating over the lockdowns.
Loss of social capital	Young people grew distant from their peripheral social network yet strengthened their relationship with inner friendship circles.
Worsening mental health	All participants reported a deterioration of their mental wellbeing over the lockdowns, but few sought appropriate support.

Main themes emerging from the fifteen qualitative interviews, briefly summarised.

## Data Availability

Data available upon request.

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
