# Peer review of "‘It’s Easily the Lowest I’ve Ever, Ever Got to’: A Qualitative Study of Young Adults’ Social Isolation during the COVID-19 Lockdowns in the UK"

_ijerph, 2021, doi:10.3390/ijerph182211777_

Round 1

Reviewer 1 Report

Thank you for the opportunity of reading and reviewing your manuscript. It addresses a topic which is under the journal's scope and certainly highly important in the current context. However, it is significantly underdeveloped and needs further work both for conceptualization, literature and for research itself.  For example, the theoretical / literature support is weak, there are no hypotheses and literature adequate for them, the structure of the paper needs adjustment (Introduction and Literature should be massively revised), the research is very weak and based on several interviews which relevance is limited and uncertain, the discussion is weak, the conclusions are quite limited. In conclusion, I consider the authors have much work to do before the paper become publishable in a top-tier journal. Good luck! 

Reviewer 2 Report

I really enjoyed reading this study.

It was very well written. I appreciated reading something with short sentences that were clear and easy to grasp.

I liked the qualitative approach quite a lot, though, as the limitations note, there were some issues that I bear mentioning. The N (15 respondents) is small even given that it is a qualitative study. I also felt like the respondents did not represent a broad enough age range (15-24 is the WHO definition of youth, but the youngest respondent was 18 and most were closer to 24 than 15). Additionally, while the sample gathering was described as broad, the respondents seemed to center around students/people in the education sphere which was a bit strange. Two were part-time grocery workers. I felt like a larger sample could've been gathered more intentionally by simply creating a convenience sample of students. The sample gathered wasn't necessarily more robust. The gender and geography issues were mentioned in the limitations. 

I like the topic and the content, but the piece is very short (which isn't necessarily a bad thing). The idea that social media/the Internet connected people to one another while also eroding mental health is a profound finding that could be elaborated upon. Social media as a group/social connector and individual/mental health disrupter could be the focus of this piece or a future one on a similar topic.

Also the idea of being a burden even to the NHS is profound. One would think that using the NHS, as a taxed benefit, would be seen as a right to some extent (or at least a service that was paid for and deserved) by citizens and yet this result counters such a view. More on that view would be interesting especially for non-UK readers who might not know how users of the NHS view the service (in other words, is it common for using NHS to be viewed as "burdening the system"? Do politicians make these claims to constituents to try to save money for other provisions?).

In all, a good and interesting study. The sample could be larger and more diverse, and perhaps that is the next step. Some of the points could be drawn out more. Important work.

Reviewer 3 Report

Thank you for conducting this well-positioned research.

Your article was well constructed and written. This is because on a broad scale there were no extraneous sections and on a finer scale the sentence structure was finely attended to. The content was logically and progressively presented.

I did not have a problem with the small sample size, in social science research it is the depth of analysis that is more important.

There was no mention of the questions asked in the semi structured Zoom interviews. Usually these are included.

To address:
Page 2 and line 82 mentions 'chicken shops' as important community spaces. This is intriguing, but not immediately identifiable, for someone like me, from another country.

On page 6 and lines 233,4 the word 'moreover' heralds you are going to add something - and you don't.

Page 6 and line 241 uses the phrase 'Befriending interventions'. This phrase needs a little explanation as during a lockdown the implication is that you are suggesting yet another online relationship. There were no previous mentions of this phrase, so you cannot just drop it into the Discussion.

Overall this was a nicely written article with an important contribution.

Reviewer 4 Report

  1. I reccomend this article for publishing.
  2. Authors should implement some studies  - /comparision with Eastern Europe /  for example: Social Media and Students"  Wellbeing : An Empirical analysis during the Covid-19 Pandemic , 2021 13(18). TKACOVA, H. et al. SustainabilitOr Pavlikova et. al. How to keep University active during Covid 19 Pandemic: , Sustainability 

Round 2

Reviewer 1 Report

Thank you for providing the revised version of your manuscript. My comments were, in general, addressed. 

Author Response

Many thanks for your response.